# Zoonotic Malaria Risk in Serra Do Mar, Atlantic Forest, Brazil

**DOI:** 10.3390/microorganisms11102465

**Published:** 2023-09-30

**Authors:** Marina E. O. Rangel, Ana Maria R. C. Duarte, Tatiane M. P. Oliveira, Luis F. Mucci, Ana Carolina Loss, Jose R. Loaiza, Gabriel Z. Laporta, Maria Anice M. Sallum

**Affiliations:** 1Department of Epidemiology, School of Public Health, University of São Paulo, São Paulo 01246-904, SP, Brazil; 2Laboratory of Protozoology, Institute of Tropical Medicine, School of Medicine, University of São Paulo, São Paulo 05403-000, SP, Brazil; 3Institute Pasteur, State Secretary of Health of São Paulo, São Paulo 01311-000, SP, Brazil; 4Graduate Program in Biological Sciences, Federal University of Espírito Santo, Vitória 29075-710, ES, Brazil; carol.loss@gmail.com; 5Institute of Scientific Research and High Technology Services of Panama (INDICASAT AIP), Panamá 0843-01103, Panama; 6Graduate Program in Health Sciences, FMABC University Center, Santo André 09060-870, SP, Brazil

**Keywords:** animals, *Anopheles*, molecular sequence data, protozoan proteins, sequence analysis

## Abstract

Here, the main goal is to assess natural infections of *Plasmodium* spp. in anophelines in a forest reserve from the same region where we previously found a surprisingly high rate (5.2%) of plasmodia infections (*n* = 25) in *Kerteszia* mosquitoes (*N* = 480) on the slopes of Serra do Mar, Atlantic Forest, Brazil. The mosquito collection sampling was carried out at the Legado das Águas Forest Reserve using CDC light traps and Shannon traps at night (5–10 pm) in 3-day collections in November 2021 and March, April, May, and November 2022. The captured specimens were morphologically identified at the species level and had their genomic DNA extracted in pools of up to 10 mosquitoes/pool. Each pool was tested using *18S* qPCR and *cytb* nested PCR plus sequencing. A total of 5301 mosquitoes, mostly belonging to the genus *Kerteszia* (99.7%), were sampled and sorted into 773 pools. Eight pools positive for *Plasmodium* spp. were identified: four for *Plasmodium* spp., one for *P. vivax* or *P. simium*, one for *P. malariae* or *P. brasilianum*, and two for the *P. falciparum*-like parasite. After Sanger sequencing, two results were further confirmed: *P. vivax* or *P. simium* and *P. malariae* or *P. brasilianum*. The minimum infection rate for *Kerteszia* mosquitoes was 0.15% (eight positive pools/5285 *Kerteszia* mosquitoes). The study reveals a lower-than-expected natural infection rate (expected = 5.2% vs. observed = 0.15%). This low rate relates to the absence of *Alouatta* monkeys as the main simian malaria reservoir in the studied region. Their absence was due to a significant population decline following the reemergence of yellow fever virus outbreaks in the Atlantic Forest from 2016 to 2019. However, this also indicates the existence of alternative reservoirs to infect *Kerteszia* mosquitoes. The found zoonotic species of *Plasmodium*, including the *P*. *falciparum*-like parasite, may represent a simian malaria risk and thus a challenge for malaria elimination in Brazil.

## 1. Introduction

Outside the Amazon region, *Plasmodium* transmission mainly occurs in forest remnants across the Brazilian Atlantic Forest. In November 2012, we found *Kerteszia* and *Nyssorhynchus* mosquitoes infected with a forested *Plasmodium falciparum*-like parasite on the slopes of Serra do Mar, Brazilian Atlantic Forest [1]. These samples were further tested in two other laboratories. An assay amplifying a cytochrome b (*cytb*) fragment carried out in Brazil further confirmed the *P*. *falciparum*-like parasite identified in the mosquitoes *Nyssorhynchus strodei* and *Kerteszia cruzii* sampled in Serra do Mar in 2012 [1,2]. A USA-based laboratory, which participates in the WHO testing program to evaluate the performance of labs performing the diagnosis of malaria, further confirmed the *P*. *falciparum*-like parasite in *Ke*. *cruzii* (sample *M267*) from Tapiraí County (Serra do Mar) using *18S* rRNA gene and apical membrane antigen-1 (*AMA-1*) gene qPCR assays (Appendix A). The *AMA-1* primer and probe set used in the latter assay was specific to *P*. *falciparum*; only the African apes *Plasmodium reichnowi* and *Plasmodium gaboni* could cross-react, both of which are closely related to *P*. *falciparum* (Appendix A). This prompted us to develop a hypothesis on the forested *P*. *falciparum*-like parasite in Serra do Mar, which is either an isolated emergence having genetic similarity with the African parasite or the product of an evolutionary event known as the spillover–spillback mechanism [2,3]. This mechanism seems to be subjacent with the high prevalence of *Plasmodium simium* and *Plasmodium brasilianum* among platyrrhine (New World) simians in South and Central America [4,5,6]. Both *P*. *simium* and *P. brasilianum*—which are genetically indistinguishable from *Plasmodium vivax* and *Plasmodium malariae*—are zoonotic plasmodia that cause simian malaria in humans in Brazil [3,7,8,9,10]. Clinical infections of zoonotic *Plasmodium* spp. in the Atlantic Forest can be as severe as human malaria [10], particularly if they go undiagnosed.

Zoonotic infections are accidental transmissions of plasmodia to humans from simian reservoirs in the Atlantic Forest [11,12]. *Plasmodium brasilianum* was experimentally shown to infect human volunteers after a caged colony of *Anopheles freeborni* had contact with the infected simian reservoir *Ateles geoffroyi* in Panama [13]. Indigenous people naturally carry *P*. *brasilianum* in the Yanomami Indigenous Reserve, Roraima State, Brazil [14]. *Plasmodium simium* was first found in a reported human infection in São Paulo, Brazil in the 1960s [15]. More recently, it has been discovered that the majority of clinical non-imported malaria cases are of simian origin, particularly due to *P*. *simium* in the Atlantic Forest [10]. These plasmodia are transmitted inside the forest by sylvatic anophelines, mainly *Kerteszia cruzii* [16]. This abundant vector species depends on canopy bromeliads (Bromeliaceae) to develop in the immature stages [11]. *Kerteszia* females are opportunistic and can seek non-human primates, birds, and other animals in the canopy as well as humans and other mammals at ground level [16]. Mathematical modeling has shown that simian malaria transmission to humans is more likely to occur at forest edges, where vector displacement from the canopy is greater, simian species are present, and biodiversity effects that regulate vector abundance are lower [17].

Both *P. simium* and *P. brasilianum* could result from a spillover–spillback process, in which malaria-infected humans introduced Old World lineages of *P. vivax* and *P. malariae* into native non-human primates [4,5]. After local speciation, they became zoonotic but remained genetically indistinguishable from their human malaria counterparts [2,4,5,6]. The mitochondrial lineages of *P. vivax* and *P. simium* have been found in non-human primates and humans in the Atlantic Forest, but not in the Amazon, which suggests a recent human-to-monkey transfer [18]. This is supported by the low diversity levels of *P. simium* lineages in New World monkeys compared with *P. vivax* lineages from humans living in Brazil [18]. New World *P*. *simium* and *P*. *vivax* PVS47 gamete surface protein sequences cluster together but are non-convergent with those from the Old World, probably due to local vector selection and speciation [4]. *Plasmodium brasilianum* and *P*. *malariae* present identical *18S* gene sequences, but they share human and non-human hosts in South America and Africa [6,14,19]. This further demonstrates their lack of genomic distinction but considerably high host-transfer plasticity [6,19]. The evolutionary origins of *P. malariae* and *P. brasilianum* remain disputed [20].

The epidemiological importance of *Ke. cruzii* acting as a malaria vector in Brazil dates back to the 1940s [21]. Clinical cases were attributed to its high vectorial capacity for human malarias [21]. As clinical cases became more sparsely distributed in the 1980s, zoonotic malaria fitted as an alternative hypothesis [11,22]. In 2015–2016, it was suggested that every human *vivax* malaria case locally acquired in the Atlantic Forest may be due to a *P*. *simium* zoonotic infection [10]. More recently, it was shown that New World *Alouatta* simians (platyrrhine howler monkeys) can contain sufficient *P*. *simium* gametocyte density in the blood to infect mosquitoes [23]. In addition, reports showed that *18S* and *cytb* gene sequences of *P*. *falciparum* in *Kerteszia* mosquitoes [1,2] and seroprevalence against *P*. *falciparum* circumsporozoite proteins in howler and other New World monkeys [24] posed a further zoonotic malaria risk due to *Plasmodium* spp. in the Atlantic Forest.

Here, the main goal was to analyze plasmodial natural infection in anophelines in a private and well-preserved forest reserve on the slopes of Serra do Mar in the Brazilian Atlantic Forest from 2021 to 2022. In 2012, we found a surprisingly high infection rate (5.2%) of zoonotic malarias among *Kerteszia* mosquitoes [1]. As we knew in advance that the reemergence of the yellow fever virus in Brazil from 2016 to 2019 decimated most of the *Alouatta* populations throughout the Atlantic Forest [25,26], the study hypothesis was that the 2021–2022 zoonotic malaria infection rate in *Kerteszia* should be zero (0%) as supported by the absence of *Alouatta* monkeys [25,26] to locally infect mosquitoes [23]. Alternatively, if this zoonotic malaria risk was in any estimate greater than zero, other simian reservoir species might be occurring, or human hosts may sustain human-to-human zoonotic malaria transmission in the studied area.

## 2. Materials and Methods

### 2.1. Study Area and Sampling

The Legado das Águas Forest Reserve (Water Legacy – Votorantim Reserve) is the largest private forest reserve in Brazil (31 thousand hectares) and together with other governmental forest reserves comprise the Atlantic Forest Biosphere Reserve (Figure 1). This reserve undertakes programs of ecotourism, forest conservation, wildlife monitoring, native plant nursing, scientific research, training and education, and citizen science (the One Health program). Altogether, this reserve has received >10 awards from different sources recognizing its contribution to the mission of the national conservation of the Atlantic rainforests.

A total of 13 locally acquired malaria cases in humans were officially reported from 2007 to 2022 [27]. These cases were not related to imported malaria; their causal interpretation was zoonotic malaria. In November 2012, we carried out a mosquito collection sampling near the Alecrim Waterfall (Figure 1) and found 22 plasmodium-infected *Ke. cruzii* out of 207 captured (10.6%) [1]. In December 2018, a researcher from a medical school was leading nocturnal collections of sand flies on the Cambuci Suspended Trail (Figure 1) and acquired malaria. This case was officially reported as *P. vivax* in the state of São Paulo [27]. In January 2019, we captured and tested 1536 *Ke. cruzii* from the Alecrim Waterfall and Cambuci Suspended Trail sites (Figure 1). These samples were tested in the previously mentioned USA-based laboratory, but all the tests were negative for *Plasmodium* spp. (Appendix A). In March 2022, a visitor who was participating in ecotourist activities on the December Waterfall Trail (Figure 1) was diagnosed with malaria. This case was officially reported as *P. vivax* in the state of São Paulo [27].

In the present work, we undertook a comprehensive sampling to further test mosquito infectivity for the presence of *Plasmodium* spp. in three sites in the Legado das Águas Forest Reserve (Figure 1). The December Waterfall Trail is a 3 km forested trail along with natural pools, rivers, and a 15 m high waterfall. The Cambuci Suspended Trail is a 2 km forested trail over a suspended wooden structure on a steep slope. The Tapir Creek Village is a 100-year-old local community composed of small houses surrounded by a forest and inhabited by 15–20 native residents. Mosquito sampling was carried out in 3-day collections (day 1, Cambuci Suspended Trail; day 2, December Waterfall Trail; and day 3, Tapir Creek Village) at 5–10 pm with 3 Shannon traps/day and 3 canopy CDC light traps/day in 5 periods (November 2021 and March, April, May, and November 2022). The collection period of August–September 2021 was a pilot study when we tested collection methods (Shannon and CDC traps), sampled mosquito fauna from the Cambuci Suspended Trail, and visited other sites which were not considered herein.

Mosquito species were determined based on female adult morphology using standard taxonomic keys [28] based on the nomenclature found in [29]. After species identifications, female specimens were separated into pools in microtubes that were labeled with information from field work as collection site, collection method, collection time, collection day, and so on. Storage was provided to these samples in −20 °C freezers at the University of São Paulo.

### 2.2. DNA Extraction

Species-identified females were separated into pools of up to 10 specimens/pool in 1.5 mL tubes and stored at −20 °C until DNA extraction. The only exception was during the third collection period (March) because we knew in advance that a case of malaria had been identified from a visitor who was in the studied region during Carnival (26–28 February and 1 March). We decided to individually test mosquitoes sampled in March to increase the chances of *Plasmodium* spp. detection and sequencing.

The DNA extraction from pools of whole mosquitoes was implemented using a salting-out method [1]. Details regarding this method can be found elsewhere [1]. This approach was carried out at the School of Public Health of the University of São Paulo.

The extracted DNA was stored (−20 °C) until the real-time qPCR of *18S* and conventional PCR of *cytb* plus Sanger sequencing were accomplished. These approaches were undertaken at the Institute of Tropical Medicine of the University of São Paulo.

### 2.3. Real-Time qPCR Assay

The extracted DNA was first cleaned with RNase enzyme treatment and then tested in real-time with *18S* qPCR amplifying a fragment of 157–165 bp [30]. We used a Mix TaqMan Universal (Thermofisher Scientific^™^, Waltham, MA, USA) and genus- and species-specific primers and probes for *Plasmodium* spp., *P. vivax* or *P*. *simium*, *P. malariae* or *P*. *brasilianum*, and *P. falciparum* [31]. In each testing, we had 3 μL of DNA (0.3–60 ng/UL), a primer concentration of 0.15 μM and a probe of 0.05 μM, followed by qPCR conditions: 50 °C for 2 min and 95 °C for 10 min plus 50 amplification cycles (95 °C for 15 s and 60 °C for 1 min). Positive controls were *P*. *vivax*, *P*. *falciparum*, or *P*. *malariae* DNA. Negative control was ultrapure water.

### 2.4. Nested PCR plus Sequencing

Positive samples in the real-time qPCR samples were further analyzed with a nested PCR assay amplifying a ∼402 bp *cytb* fragment [32,33]. The first reaction was carried out with a DreamTaq PCR Master Mix (Thermofisher Scientific™, Waltham, MA, USA). We used 5 µL of DNA and 1 µL of each primer (Pf_F3700 and Pf_R4615), resulting in a 915 bp fragment [32,33]. For the second reaction, we used a DreamTaq PCR Master Mix (Thermofisher Scientific™, Waltham, MA, USA) with the 4 µL PCR product from the first reaction and 1 µL for each primer (Pf_F3700 and Pf_R4615), resulting in a ∼402 bp fragment [32]. Conditions for both assays were as follows: 1 cycle of denaturation at 94 °C for 5 min, 30 cycles of denaturation at 94 °C for 15 s, and annealing at 55 °C (the primers’ first pair) or 52 °C (the primers’ second pair) for 15 s with a final extension for 2–5 min at 72 °C. PCR products were observed in agarose gel (Sigma-Aldrich^™^, Saint Louis, MO, USA) with a concentration of 3.0% in 1x TBE buffer (0.09 M Tris-borate, 0.002 M EDTA). A 50 bp molecular weight marker was used and the gel was visualized under ultraviolet light. Fragments having the expected sizes (∼402 bp) were forwarded to a Sanger sequencing company (Genomic Engenharia Molecular, São Paulo, SP, Brazil) [2].

### 2.5. Phylogenetic Analyses

The *cytb* sequences generated herein were compared with databases available using the BLAST tool (Basic Local Alignment Search Tool; https://blast.ncbi.nlm.nih.gov, accessed on 29 September 2023) on GenBank (https://www.ncbi.nlm.nih.gov/genbank/, accessed on 29 September 2023). These *cytb* sequences were aligned on the Geneious 6.1 program (Biomatters, Auckland, New Zealand) along with 16 other *Plasmodium* sequences available from GenBank, including *Plasmodium gallinaceum* and *Plasmodium relictum* as outgroups (~363 bp). Phylogenetic reconstruction with maximum likelihood and bootstrapping in the RAxML program on the CIPRES platform (https://www.phylo.org/portal2, accessed on 29 September 2023) was implemented. GTR+G was the molecular evolution model assumed.

### 2.6. Data Analysis

The minimum infection rate (MIR) was estimated as follows: MIR = $\frac{x}{n}$ × 100, where *x* is the number of positive pools and *n* is the total number of mosquitoes tested. This approach has been used previously [34]. The main assumption was that a positive pool had at least one *Plasmodium*-infected mosquito. For the collections in the third period (March), any positive result meant a natural infection in a single mosquito.

## 3. Results

### 3.1. Sampled and Tested Specimens

A total of 5301 anophelines were captured from the selected sampling sites (Figure 1). A total of 80 h of sampling efforts (3 days/site × 5 h × 5 periods = 75 h + 5 h in the pilot study/one site) resulted in an average of 66 anophelines captured per hour. The majority were *Ke. cruzii* (*n* = 5133; 97%) or *Kerteszia* mosquitoes (*n* = 5285; 99.7%) and a few belonged to other species (*An*. *lutzii*, *An*. *medialis*, and *An*. *fluminensis*). Shannon traps were the most efficient (5226 captured anophelines), while CDC light traps captured 75 anophelines (Table 1). We sampled 108 mosquitoes at the Cambuci Suspended Trail site from August to September 2021, which was a pilot study collection. From November 2021 on, we collected samples at all three collection sites. The number of mosquitoes sampled varied as follows: 2746 in November 2021, 184 in March, 1209 in April, 162 in May, and 892 in November 2022.

### 3.2. Real-Time qPCR Assay

Out of the 773 pools tested, real-time qPCR resulted in eight positive pools (1%) in *Kerteszia* females (Table 1). In March, we individually tested all mosquitoes because we knew a human malaria case was identified from a visitor who had been hiking on the December Waterfall Trail one week before our field collections from March 8th to 10th. One sample (D_3210; one *Ke. cruzii* specimen) was *18S*-positive for *P. vivax* or *P. simium*, showing a cycle threshold of (ct) = 40. This malaria-positive mosquito female was collected in a Shannon trap at 19–20 h (Table 1). In April, five pools were positive (Table 1). One of them was *18S*-positive for *P. falciparum*, also known as the forested *P. falciparum*-like parasite (ct = 40). This was obtained from a pool (S_3723) with two *Ke. cruzii* females captured in Shannon traps at 20–21 h at the Cambuci Suspended Trail site (Table 1). Another one was *18S*-positive for *P. malariae* or *P. brasilianum* (ct = 47) from a pool (RA_4518) with three *Ke. cruzii* females captured in Shannon traps at 20–21 h at the Tapir Creek Village site (Table 1). In November 2022, we had one *18S*-positive sample for *P. falciparum* (also known as the forested *P. falciparum*-like parasite; ct = 47) in a pool (D_4972) of two *Ke. cruzii* females captured in Shannon traps at 17–18 h at the December Waterfall Trail site (Table 1). The remaining four were *18S*-positive for plasmodium-genus only in pools with 10 *Ke. cruzii* female specimens (RA_4366 ct = 35, RA_4397 ct = 31, and RA_4457 ct = 45) in April and (RA_5166 ct = 44) in November 2022 at the Tapir Creek Village site (Table 1).

### 3.3. Nested PCR plus Sequencing and Phylogenetic Analyses

Two *18S*-qPCR-positive pools were further confirmed with nested PCR plus sequencing (Table 1). The *P. vivax* or *P. simium* (GenBank ID OQ830680; Table 1) nucleotide sequences herein obtained were 98.1% similar to the *P. vivax* mitogenome from human samples in Brazil [18] (Figure 2) and elsewhere in Papua New Guinea and Panama. The *P. malariae* or *P. brasilianum* (GenBank ID OQ830681; Table 1) nucleotide sequences herein obtained were 99% similar to the *P. malariae* mitogenome in feces of *Pan troglodytes* (GenBank ID MF693452) and *Gorilla* (GenBank ID MF693444 and MF693446) and from a human sample (GenBank ID MF693442) in Cameroon, Central Africa [35]. The sample also showed 99% similarity with *P. brasilianum* collected from callitrichid monkeys (*Saguinus fuscicollis* and *Saguinus imperator*) in Madre de Dios, Southeastern Peru (GenBank ID MF693425) [36] (Figure 2).

### 3.4. Data Analysis

The minimum infection rates in *Kerteszia* females are shown in Table 2. These are global estimates. If we considered only the sampled mosquitoes in the day when one was found positive, the estimates would be much higher. For instance, we collected a total of 57 *Ke. cruzii* females in a one-day collection in March at the December Waterfall Trail site when a *P. simium*-infected female was identified. For this day and site, the infection rate was 1.75% (1/57) or 17× higher than the global estimate of 0.1% (Table 2). During this month, a human malaria case was reported from an ecotourist who engaged in forest activities at this site.

## 4. Discussion

The zoonotic malaria risk estimated as the minimum infection rate in *Kerteszia* females with *Plasmodium* spp. ranged from 0.1 to 0.2% in the Legado das Águas Forest Reserve from 2021 to 2022. This means that the zoonotic malaria risk for visitors or staff engaging in forest activities is as low as one–two plasmodium-infected persons per 1000 human-seeking *Kerteszia* females. This risk is notably lower than that obtained 10 years ago (5–10%) [1]. On one hand, this outcome was in line with the study’s hypothesis, i.e., with the decline of howler monkeys as the main reservoir in the studied area, the zoonotic malaria risk dropped to low levels [25]. On the other hand, an alternative simian host to infect *Ke. cruzii* with *P. simium*, *P. brasilianum*, and the *P. falciparum*-like parasite was a puzzling outcome and needs further comments, as seen below.

Before proceeding to explain this alternative hypothesis, it is important to highlight the difference between global estimates vs. day/site estimates when interpreting our outcomes. We identified an anopheline infection rate 17× higher than the global estimate (1.75 vs. 0.1%) on 9th March 2022 at the December Waterfall Trail site. This collection occurred one week after a visitor participated in activities at this site and contracted malaria. This case, officially reported as *P. vivax*, was diagnosed by blood smear slides in a reference center for malaria in São Paulo [27]. Here, we obtained an *18S* fragment sequence 98% identical to *P. vivax* or *P. simium* [18] at the same site and period from a *Kerteszia* female. In other words, alternative reservoirs as evidenced below might be able to sustain highly dynamic transmission and support a strong force of infection of simian parasites to humans, conditioned to time and space.

Atlantic Forest howler monkeys (*Alouatta guariba clamitans*) are the main reservoir for simian malaria [7,23,37]. However, their populations steeply declined after yellow fever virus outbreaks from 2016 to 2017 [25,38]. The Legado das Águas Forest Reserve has 20–30 forest rangers who walk 10–15 km on a daily basis to prevent trespassing from wildlife hunters or palm tree smugglers. While doing this, they make important observations on currently present wildlife in this reserve. These rangers confirmed the absence of howler monkey sightings between 2018 and 2022. This suggests that other simian species can harbor *Plasmodium* spp. and infect *Kerteszia* in the absence of howler monkeys. In agreement with this view is another study undertaken on the slopes of Serra do Mar in Espírito Santo State where anopheline infection rates were unaffected by yellow fever outbreaks and local *Alouatta* population decline [39]. We hypothesize that one or two of the remaining simian species in the Legado das Águas Forest Reserve can play a role as alternative reservoirs. Southern Muriqui (*Brachyteles arachnoides*) and capuchin monkeys (*Sapajus*) are frequently seen by rangers in forested areas from this reserve and both have been previously found to be infected with *P. simium* and *P. brasilianum* [40,41,42]. During this research, we trained forest rangers to obtain feces samples from these simians during their daily monitoring (Figure 3). These simian fecal samples (*n* = 12) were tested and one of them from a Southern Muriqui was positive (ct = 45) for plasmodium-genus only (Figure 3).

The main limitation faced by this study was lower levels of specificity for tests carried out with non-human samples (mosquitoes) conditioned to complexities of *Plasmodium* evolutionary lineages in the Atlantic Forest.

## 5. Conclusions

Naturally plasmodium-infected *Kerteszia cruzii* were found in Serra do Mar, Atlantic Forest at ground level and in the absence of their main reservoir (*Alouatta* monkeys), which suggests the role of Southern Muriqui (*Brachyteles arachnoides*) as alternative reservoirs.

We found potential agents of zoonotic simian malaria, also known as *P. simium* and the *P. falciparum*-like parasite, on the December Waterfall Trail, *P. brasilianum* and four unknown *Plasmodium* species in the Tapir Creek Village, and *P. falciparum*-like on the Cambuci Suspended Trail.

## Figures and Tables

**Figure 1 microorganisms-11-02465-f001:**
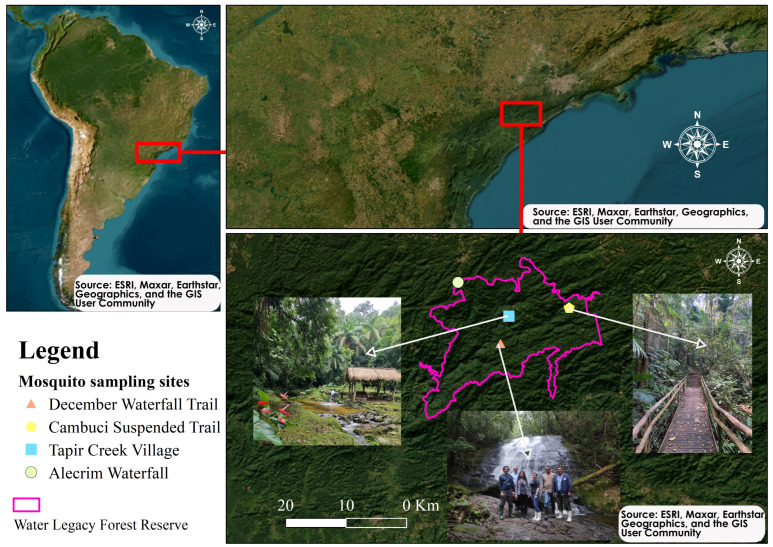
Study area and sampling sites. In 2012, we found a surprisingly high *Plasmodium* spp. infectivity rate among *Kerteszia* mosquitoes at the Alecrim Waterfall site. In the present work, we tested the infectivity of sampled mosquitoes at three sites in the Legado das Águas Forest Reserve: (1) December Waterfall Trail, (2) Cambuci Suspended Trail, and (3) Tapir Creek Village.

**Figure 2 microorganisms-11-02465-f002:**
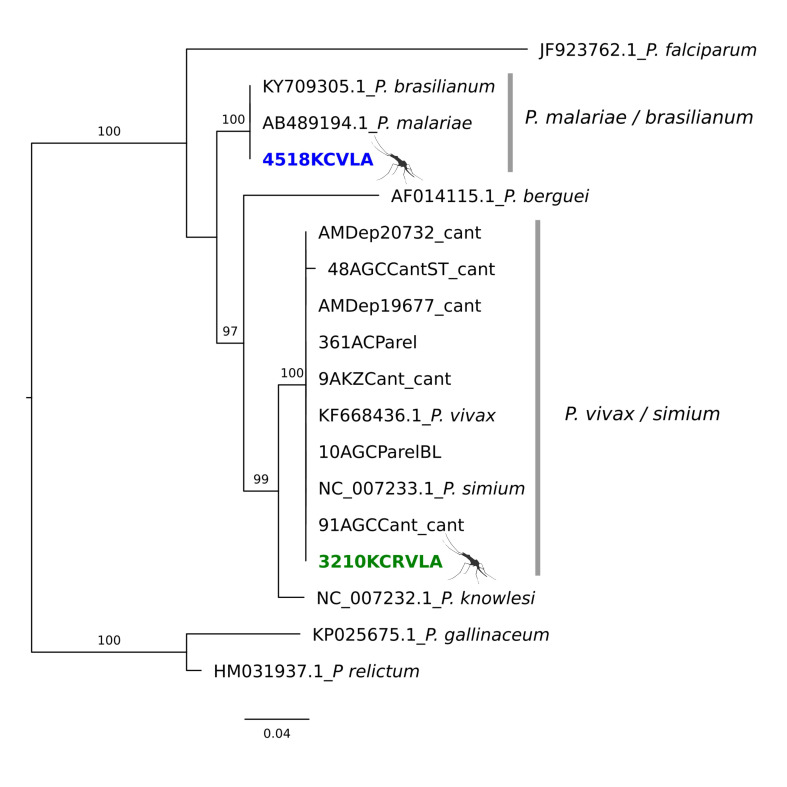
Maximum-likelihood phylogenetic tree of *Plasmodium* species including the two sequences obtained from *Ke. cruzii* naturally infected in the Legado das Águas Forest Reserve, São Paulo, Brazil (in bold). The values in the branches are bootstraps. Blue (GenBank ID OQ830681) and green (GenBank ID OQ830680) colors are the samples obtained herein.

**Figure 3 microorganisms-11-02465-f003:**
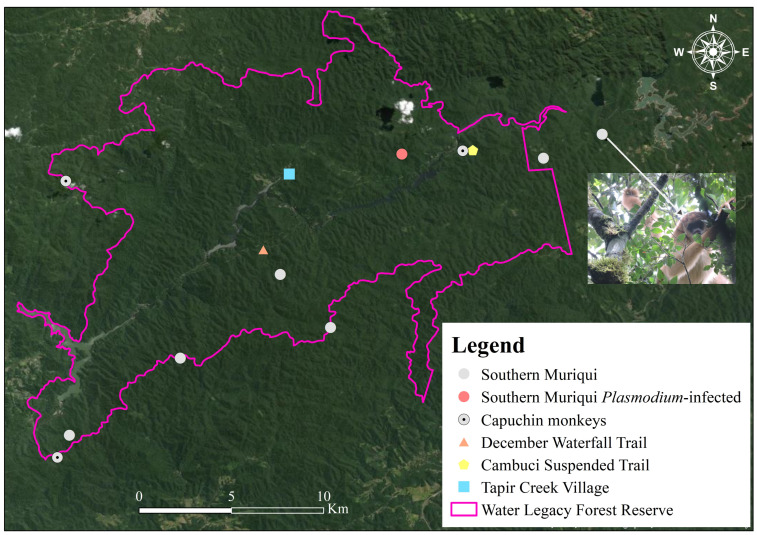
Locations with exact geographic coordinates of simian fecal samples obtained from forest rangers for this project. One fecal sample from a Southern Muriqui tested positive with *18S* qPCR for *Plasmodium* spp. (ct = 45).

**Table 1 microorganisms-11-02465-t001:** Anopheline females captured at Legado das Águas Forest Reserve, 2021–2022.

Sampling Site	Anophelines (*n*)	Pool Positive for *Plasmodium* spp. with *18S* qPCR ^1^			
		*Plasmodium*	*P. vivax*	*P. falciparum*	*P. malariae*
Cambuci Suspended Trail	*Ke. cruzii* (853) *Kerteszia* spp. (2) *An. lutzii* (3) *An. medialis* (1)	- - - -	- - - -	1 (April 22) - - -	- - - -
December Waterfall Trail	*Ke. cruzii* (1928) *Kerteszia* spp. (114)	- -	1 (March 22) ^2^ -	1 (November 22) -	- -
Tapir Creek Village	*Ke. cruzii* (2352) *Kerteszia* spp. (36) *An. lutzii* (3) *An. medialis* (5) *An. fluminensis* (4)	3 (April 22) 1 (November 22) - - -	- - - - -	- - - - -	1 (April 22) ^3^ - - - -

^1^ Pool containing 2–10 specimens, except for collections in March when mosquitoes were individually tested. ^2^
*P. vivax* or *P. simium* sequenced with *18S* qPCR and *cytb* 402 bp (GenBank ID OQ830680). ^3^
*P. malariae* or *P. brasilianum* sequenced with *18S* qPCR and *cytb* 402 bp (GenBank ID OQ830681).

**Table 2 microorganisms-11-02465-t002:** Minimum infection rate in *Kerteszia* females to estimate zoonotic malaria risk per selected sampling site in the Legado das Águas Forest Reserve, 2021–2022.

Sampling Site	*Plasmodium*	*P. malariae*	*P. vivax*	*P. falciparum*
Cambuci Suspended Trail	-	-	-	$\frac{1}{855}$ = 0.1%
December Waterfall Trail	-	-	$\frac{2}{2042}$ = 0.1%	
Tapir Creek Village	$\frac{5}{2388}$ = 0.2%		-	-

## Data Availability

Data are contained within the article.

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
