# Peer review of "Zoonotic Malaria Risk in Serra Do Mar, Atlantic Forest, Brazil"

_microorganisms, 2023, doi:10.3390/microorganisms11102465_

Round 1

Reviewer 1 Report

This is a very interesting and original study to assess natural infection by Plasmodium spp. in Kertezia mosquitoes in Serra do Mar, Atlantic Forest, Brazil, where the yellow fever virus decimated most of the Alouatta population in previous years. 

The results of the present study are very interesting since the authors detected malaria infection in mosquitoes and exposed the possibility of a new species of non-human primate different from Alouatta monkeys as a host for malaria in the study area.

Author Response

We’d like to thank the reviewer for the kind comments made on our manuscript.

Based on them, we retained most part (>95%) of the original submitted version intact after this first round of review.

Reviewer 2 Report

All required changes are on the attached revised PDF document 

Author Response

Thank you very much for considering your time to suggest corrections on the text of our manuscript. All corrections were accepted and are merged into the re-submitted version.

Additionally, we want to inform that sequences deposited in the GenBank will be available on 23rd September 2023. Please find attached further information about them.

Reviewer 3 Report

This manuscript from Rangel et al details detection and sequencing of Plasmodium species being carried by different mosquito species in the Brazilian Atlantic forest.  This was achieved by pooling collected mosquitos and testing for the presence of Plasmodium DNA with real-time RT-qPCR.  The authors noted a significantly lower presence of Plasmodium in Ke. cruzii mosquitos than a previous study from 2012, attributing this drop to the drop in Howler monkey populations as a reservoir species during the interim time.  However, the continued presence of plasmodium among mosquitos in this region indicates the potential for other local primate species to act as alternative reservoirs.

The study is straightforward and the results are clearly presented.  My biggest comment to be addressed before publication is to modify the conclusion that a high incidence of plasmodium in the December Waterfall trail in March of 2022 may be correlated with a human malaria case associated with this same region and time, as this high incidence rate was obtained from a single positive insect and therefore may be subject to bias due to a small sampling size.  

Minor editing of English language needed in a few spots in the paper

Author Response

We thank the reviewer for the insightful comment and view on our manuscript.

We agree with the suggested correction in the Conclusions section. Accordingly, we excluded the following sentence in the last paragraph of Conclusions: “Given time and space may yield higher-than-expected anopheline infections rates by simian parasites which may confer further risks to humans.”. Now, only between us and the reviewer, we truly believe that simian malaria force of infection is highly dynamic and conditioned to time (date of collection) and space (site of collection) throughout Atlantic Forest biome. But, as the reviewer noted, our sampling is robust for the estimation of global estimates, but not for site-specific estimates; therefore, it is not adequate to make conclusions or big statements from site-specific estimates. Notwithstanding, one site-specific estimate is still presented in Results and interpreted in the Discussion.

We’ve checked English writing and corrected some mistakes. It can be further improved.